# Single particles as resonators for thermomechanical analysis

Peter Ouma Okeyo[1,2,3 ✉], Peter Emil Larsen[2,3], Eric Ofosu Kissi [4], Fatemeh Ajalloueian[2,3], Thomas Rades[1], Jukka Rantanen[1] & Anja Boisen [2,3 ✉]

Thermal methods are indispensable for the characterization of most materials. However, the existing methods require bulk amounts for analysis and give an averaged response of a material. This can be especially challenging in a biomedical setting, where only very limited amounts of material are initially available. Nano- and microelectromechanical systems (NEMS/MEMS) offer the possibility of conducting thermal analysis on small amounts of materials in the nano-microgram range, but cleanroom fabricated resonators are required. Here, we report the use of single drug and collagen particles as micro mechanical resonators, thereby eliminating the need for cleanroom fabrication. Furthermore, the proposed method reveals additional thermal transitions that are undetected by standard thermal methods and provide the possibility of understanding fundamental changes in the mechanical properties of the materials during thermal cycling. This method is applicable to a variety of different materials and opens the door to fundamental mechanistic insights.

[1] Department of Pharmacy, University of Copenhagen, Universitetsparken 2, 2100 Copenhagen, Denmark. [2] Department of Health Technology, Technical University of Denmark, Ørsted Plads, 2800 Kgs. Lyngby, Denmark. [3] The Danish National Research Foundation and Villum Foundation's Center for Intelligent Drug Delivery and Sensing Using Microcontainers and Nanomechanics (IDUN), Department of Health Technology, Technical University of Denmark, Ørsted Plads, 2800 Kgs. Lyngby, Denmark. [4] Department of Pharmacy, University of Oslo, P.O.Box 1068 Blindern, 0316 Oslo, Norway. ✉email: peteoey@dtu.dk; aboi@dtu.dk

The primary goal in the early stages of materials research is to gain an understanding of the physiochemical and mechanical properties. One of the key challenges associated with the characterisation of drugs or biological matter during development is to prevent unwanted[1] phase transitions that can occur in the end-product, manufacturing or storage[2]. Therefore, controlled experiments need to be carried out in order to understand the properties of materials already in the early drug development phase. Detecting and resolving phase transitions can be difficult depending on experimental conditions. Recent studies[3,4] suggest that this is due to high energy metastable intermediates that are frequently overlooked because they appear over short time scales (~milliseconds) in different regions of single particles. They are also known to be more soluble than their stable counterparts, which is an essential property for the evaluation of the performance of drugs[3].

Standard thermal methods such as differential scanning calorimetry (DSC), thermal gravimetric analysis (TGA) and dynamic mechanical analysis (DMA)[5] are heavily used in the investigation of the physical and mechanical properties of most drugs and biological matter in early development. DSC is one of the most commonly used techniques in understanding material properties (e.g. melting point, percentage of crystallinity[6] and heat capacity/crystallisation), however, it is often unable to resolve overlapping thermal transitions at a single particle level[7]. Modulated-DSC (m-DSC)[8–10], and DMA are used for detecting subtle transitions[11], but they are limited to heating rates of 0.5–5 °C/min[12,13]. DMA works by applying an oscillating stress or strain on a sample as a function of temperature or time[14]. It induces an external pressure on the sample prior to the measurement, which is unsuitable for porous or fragile samples, such as hydrates and dehydrated materials. TGA is commonly used for water content determination[15] and can be useful for determining the dehydration pathway of hydrates, however, subtle thermal transitions, associated with metastable forms, are challenging to resolve/isolate. All these noted techniques require a minimum of several milligrams of material for analysis and give averaged responses[16–18] that makes single particle analysis simply inaccessible. It is challenging to carry out a comprehensive evaluation if there is only a limited amount of material available, or when thermal transitions need to be resolved in heterogeneous samples during early preformulation[19] stage (drugs are often not pure[11,20–22] and contain the material of interest together with impurities or different solid-state forms)[22]. This challenge may be even more prominent when investigating macromolecular structures, like proteins, that typically, in comparison to small molecules, have more complex thermal transitions, that are related to structural changes and denaturation[23].

Various groups have demonstrated the benefits of small volume analysis versus bulk analysis in early drug development[24]. For example, recently a study demonstrated the value of assessing key material properties (powder flow and tabletability) by investigating the mechanical behaviour of single particles using nanoindentation[25]. This study showed that it is possible to screen solid forms and select drugs with optimal mechanical properties with only microgram amounts of material. There is a trend in thermomechanical analysis towards smaller sample sizes using nanoelectromechanical systems (NEMS)[26] and microelectromechanical systems (MEMS), which is motivating the use of these methods in pharmaceutical research[24,27,28]. Thermomechanical characterisation of organic samples has for example been performed using string resonators and the findings show that strings are two orders of magnitude more sensitive due to the small sample mass in comparison to DSC[29,30]. This approach allowed for the detection of thermal transitions that were not detectable in DSC.

However, two of the main drawbacks using NEMS/MEMS devices for thermomechanical characterisation are the limit of access to these devices and the challenge of placing samples on the devices. Here, we report a method called 'Particle Mechanical Thermal Analysis' (PMTA) that uses a single particle as a resonator to determine changes in the mechanical properties during thermal cycling and eliminating the need for cleanroom fabrication. Thermomechanical characterisation of two model particles was performed on elongated particles of theophylline monohydrate (TP MH) and a rolled sheet of collagen particles. The proposed method can be applied to a variety of micro sized particles since most objects vibrate at certain resonance frequencies that are dependent on material and geometry[31,32].

Crystalline hydrate particles (e.g. TP MH) undergo a visual change during dehydration, from being transparent to opaque due to reorganisation of the crystal structure[33]. The rate of this phase transition can vary depending on the size[34] and dimensions of the particle, which was a motivation for investigating different sized particles with varying lengths (300–1500 μm), widths (10–150 μm) and thicknesses (~20 nm–1 μm). This transition can be monitored with thermomicroscopy and serves as an easy validation of the PMTA thermomechanical results. Multiple vibrational modes were investigated to probe spatial differences in the particles by actuating the resonators at different frequencies with a piezo crystal. Also, a correlation between the movement of the tip of a particle and its thermal transitions were studied. In addition, the denaturation mechanism of collagen particles (rattail) was investigated. This is because collagen particles have multiple medical applications, however, there is still limited knowledge on their fundamental thermomechanical response[35]. Traditionally fabricated silicon nitride resonators have well defined structural and mechanical parameters that allow for a straightforward application of the Euler-Bernouilli theory[36]. However, with organic heterogeneous structures, there are limitations[27] to the application of the theory due to defects and irregular shapes (See optical images in Supplementary Figs. 1–4, Supplementary Note 1). Even with these limitations, it is still possible to make approximations on the thermomechanical response of materials as demonstrated with TP MH and collagen fibrils. This shows the ability of this method to be used for particles that are not only shaped as beams but in principle any resonating structure with relevant applied theories.

## Results

**Theoretical considerations for single particle resonators**. There are a number of analytical models that can be used to describe mechanical structures and the changes they undergo during thermomechanical analysis. Two key parameters are used for thermomechanical characterisation of the model particles and they are; resonance frequency ($f_{res}$) and quality factor ($Q$). TP MH and collagen particles, used in this work, were treated as cantilevers and the Euler-Bernoulli beam theory was applied in order to explain the thermomechanical data[37]. Eq. (1) was used in our studies to define the $f_{res}$ of the flexural vibrational modes of cantilevers, where $\Omega$ is the eigenfrequency, $\beta_n$ is a dimensionless coefficient that is determined by the vibration mode[38], $E$ is Young's modulus, $I_{yy}$ is the second moment of inertia, $\rho$ is the density, $A$ is the cross-sectional area and $L$ is the length[37]. Thus, changes of any of these parameters during the experiments can be measured through changes in the $f_{res}$. The numerator under the square root represents the effective stiffness of the mechanical system, whereas the denominator represents the effective mass. It is assumed that the TP MH particles have a rectangular cross-section. The density of TP MH was assumed to be $1.52\,\mathrm{g\,cm^{-3}}$ and $1.49\,\mathrm{g\,cm^{-3}}$ for TP AH[39]. The collagen fibrils are assumed to

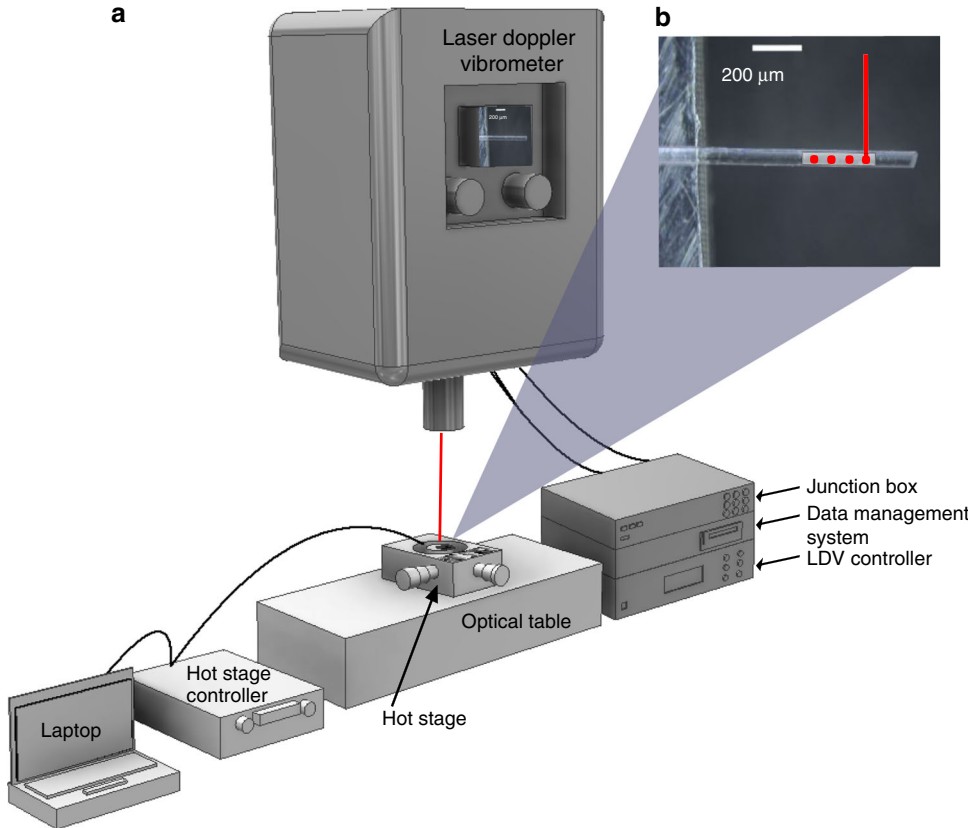

**Fig. 1 PMTA schematic and operation. a** Schematic representation of the measurement setup for PMTA where the junction box, data management system and laser doppler vibrometer (LDV) controller are used for operating and managing the data generated from the LDV. **b** Optical image of a single particle of theophylline monohydrate (TP MH) used as a cantilever with a laser spot on top (red) from the LDV as well as a measurement grid with measurement positions (red spots) that are defined using the LDV software. The particle is mounted on a 5 mm × 5 mm aluminium block (left side of the optical image). The piezoelectric crystal was placed adjacent to the aluminium block during the measurements.

have a circular cross-section. The width to height ratio of the resonators does not always fit the cantilever theory range where width/height >5[40], however, this was acceptable because the focus of our studies was to determine the main phase transitions of the proposed resonators.

$$f_{res} \approx \frac{\Omega}{2\pi} = \frac{\beta_n^2}{2\pi L^2}\sqrt{\frac{EI_{yy}}{pA}} = \sqrt{\frac{k_{eff}}{m_{eff}}} \quad (1)$$

The quality factor ($Q$) is defined in Eq 2 as a measure of the mechanical damping of the system. It is the ratio of the amount of mechanical energy stored in the system and the energy dissipated at each vibrational cycle and provides a measure of the internal viscous damping effects and friction within the investigated particles. $Q$ is extracted from the slope of the phase between the actuation of the signal and resonator response at the resonance frequency:

$$Q = -\frac{f_{res}}{2}\frac{d\phi(f)}{df}\bigg|_{f=f_{res}} \quad (2)$$

Equation 3a, b are used to make an approximation of the amount of water loss in the particles during dehydration. The relative mass change $\delta m$ can be estimated from the relative frequency change $\delta f$ according to Eq. 3b under the assumption of unchanged resonator stiffness:

$$\delta f = \frac{f_{res} - f_0}{f_0} \quad (3a)$$

$$\delta f = -\delta m = \frac{m_0 + \Delta m}{m_0} \quad (3b)$$

$\delta f =$ where $f_{res}$ is the resonance frequency, $f_0$ is the initial resonance frequency, $\Delta m$ is the mass change and $m_0$ is the initial resonator mass.

**Solid state analysis confirmation.** DSC, TGA, DMA and X-ray powder diffraction (XRPD) confirmed the solid-state form of TP MH and the collagen fibres (Supplementary Figs. 5 and 6). The main dehydration and denaturation temperatures from the DSC, TGA and DMA measurements were in agreement with PMTA.

**Single particle resonator analysis.** Figure 1 shows a schematic of the measurement setup used for PMTA (Details in Materials and method section). Single particles were placed as cantilevers on a supporting platform and subjected to controlled heating and cooling cycles (5, 20 °C/min) as well as isothermal measurements (50 °C) whilst tracking changes in their resonant behaviour with a laser Doppler vibrometer (LDV). For each experiment, a piezo-electric crystal was used for actuation of the particles during thermal cycling. A scanning grid (3–30 points, 10 ms–1 s/point) was defined based on the particle size using the microscope on the LDV (Supplementary Fig. 7).

Figure 2 illustrates a proof of concept of the PMTA method. The optical images in Fig. 2a show a major loss in the transparency of TP MH at 85 °C, which is highlighted by the vertical turquoise line.

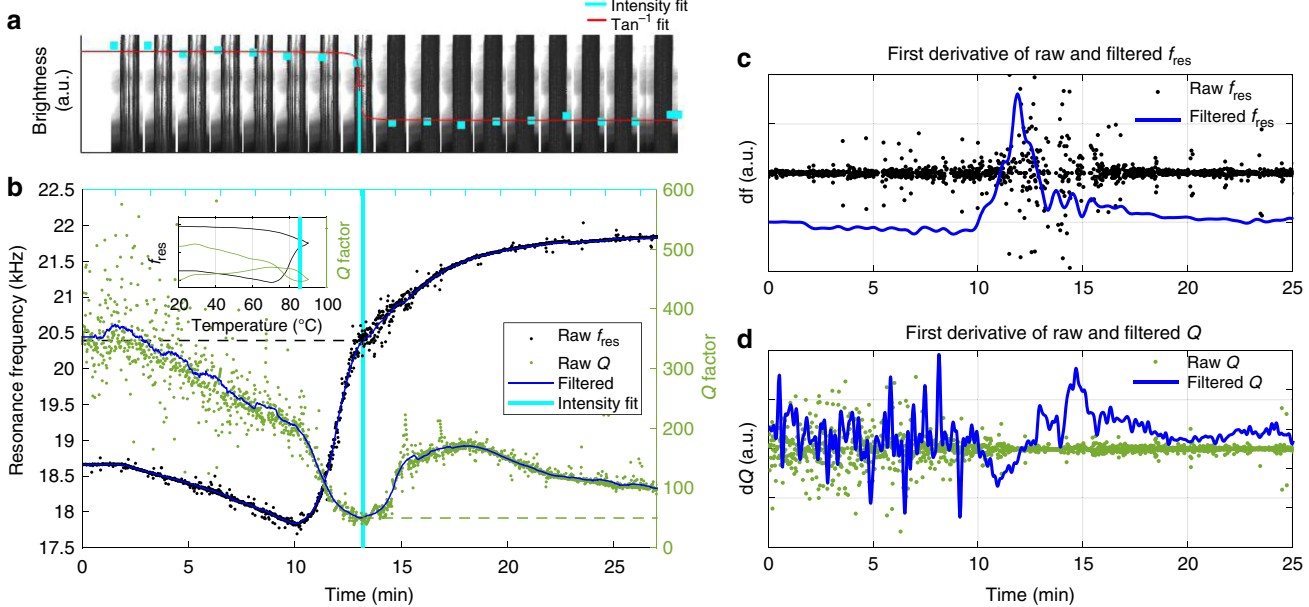

**Fig. 2 Proof of concept for PMTA from a single experiment. a** A series of optical images of a single TP MH particle (1000 μm × 65 μm, cantilever) during dehydration (25–90–25 °C, 5 °C /min) with its tracked intensity (arctan function was fitted). **b** PMTA thermogram showing the raw data for tracking of $f_{res}$ and $Q$ of TP MH during heating and cooling with the temperature plot inset. **c**, **d** First derivative plots from the tracked and filtered $f_{res}$ and $Q$ data in $b$.

From the optical images it can be seen that during dehydration, there is a progressive formation of dark regions (nucleation of anhydrous theophylline) in an anisotropic manner in the TP MH particle (Supplementary Movie 1, 2, 3, 4, 5 and 6). These transitions have previously been explained as a rearrangement of the crystal lattice. These nuclei of the anhydrous forms then form numerous crystallites, which are observed as a loss of transparency of the particle due to light scattering from the surfaces of these crystallites[33].

This observed transition in the optical images corresponds well with significant changes occurring in the particle's resonant behaviour (Fig. 2b), where the raw data and low-pass filtered results of $f_{res}$ and $Q$ (Eq. 2) are shown during a thermal cycle of TP MH (Supplementary Fig. 8 for raw data temperature plot). The PMTA thermogram (25–90–25 °C, 5 °C/min) of the TP MH particle shows a steady $f_{res}$ for the first two minutes of the experiments, thereafter $f_{res}$ starts to drop linearly. After 10 minutes there is an increase in $f_{res}$ (2.7%) that reaches a plateau after 20 min. $Q$ initially drops and starts to increase once the particle is anhydrous. At around 18 minutes, $Q$ gradually starts to drop again. The first derivative plots in Fig. 2c, d explain that the main changes occurring in $f_{res}$ are in the main phase transitions whereas with $Q$ it is from the beginning up to the 20 minute mark, which confirms the results in Fig. 2b. Considering Eq. 1, the drop in $f_{res}$ is due to the softening of the TP particle that is due to a drop in Young's modulus[41] of TP MH. The subsequent increase in $f_{res}$ is related to the stiffening (increase in Young's modulus[39]) of the particle during and after water loss. This process occurs via an anisotropic[42] mechanism due to the presence of a channel arrangement of water molecules. The multistep increase in $f_{res}$ (10–25 min) is due to the flipping of the TP MH crystal structure to the anhydrous form and this is supported by variable temperature XRD (Supplementary Fig. 9, Supplementary Note 2)[43]. The Qs in the beginning of the experiment are linked to the softening of the particle that allows for an increased mobility of water and host molecules.

It is proposed that the thermal transitions that result in sudden changes in $f_{res}$ and $Q$ are due to the presence of solid-state transitions, in particular, metastable phases of TP that eventually results in the formation of its anhydrous form[44]. These metastable phases possess different conformational arrangements[16] thereby resulting in different vibrational responses that explain the notable changes in the in $f_{res}$ and $Q$ (Supplementary Fig. 10). Revealing metastable phases is of great pharmaceutical relevance[45] as they have different physical and mechanical properties that are desirable (due to their increased solubility).

**Comparison between standard thermal methods and PMTA.** Several standard thermal methods (Fig. 3a) were compared to PMTA. One major dehydration event was observed with DSC and TGA (as shown by the highlighted areas in this figure) whereas PMTA showed this event and additional thermal transitions of TP MH related to $f_{res}$ and $Q$ (Supplementary Fig. 11 for the raw thermomechanical spectra). The DSC thermogram showed a broad endotherm (60–85 °C, 34.8 kJ/mol) and the TGA thermogram revealed a weight loss of 9% (58–72 °C) that corresponds to one mole of water per mole of the drug. The $f_{res}$ and $Q$ of another TP MH particle were tracked from 25–90–25 °C at a constant rate of 5 °C/min (Fig. 3b). The first out of plane vibrational mode was identified at 24 kHz (Q:162). $f_{res}$ drops linearly until 60 °C, and then increases in the temperature interval 61–70 °C. Based on Eq. 3a, b the mass loss was calculated to be approximately 9% which is in agreement with the TGA results. The increase in $f_{res}$, at the main dehydration of the resonator, is 3.5 kHz, which demonstrates that the dominating effect on the resonator is internal dampening due to the presence of water molecules (See Supplementary Eq. 1). $Q$ drops in the temperature range 25–45 °C, has a slight increase and an additional drop in the range 45–65 °C, recovers partially (65–78 °C) and then drops at the main dehydration event at 80 °C. The recovery of $Q$ is attributed to the progressive formation of the stable anhydrous form of TP. The $Q$ shows transitions at 45 °C and after 70 °C that remain undetected by DSC or TGA. The transition at 45 °C is a minor transition and is linked to a subtle local internal change that eventually leads to the collapse of the resonators crystal structure (See Supplementary Fig. 9b, c). At 70 °C, the resonator

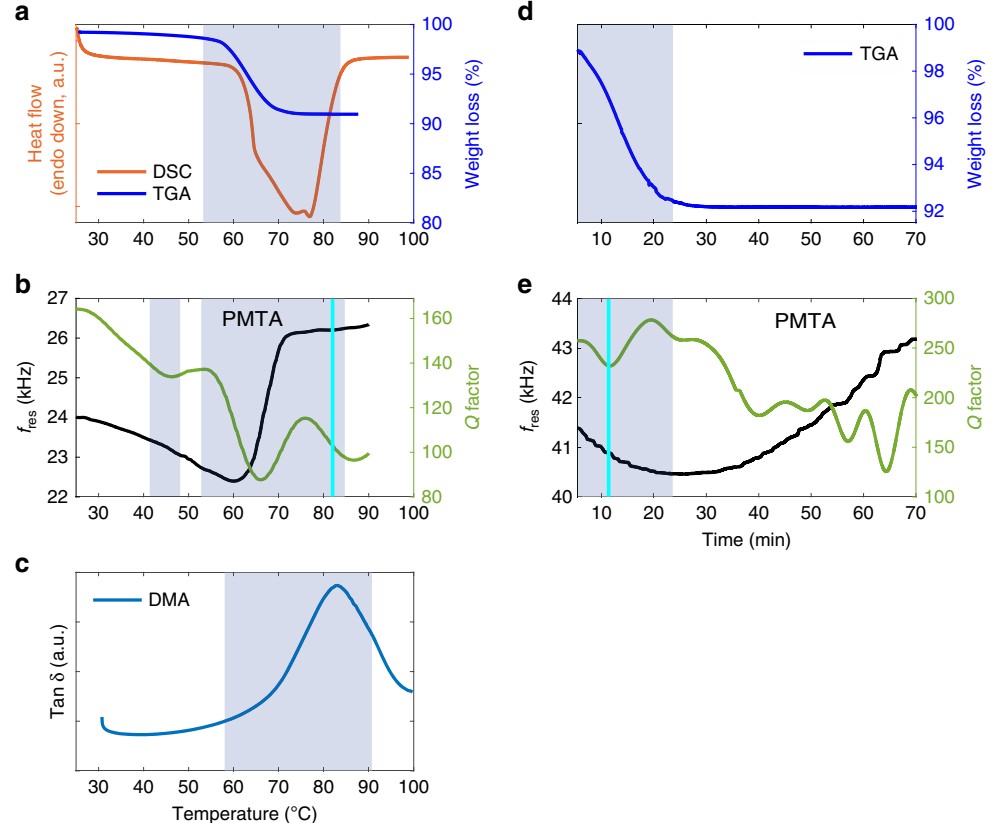

**Fig. 3 Comparison of standard thermal methods to PMTA. a** DSC and TGA thermograms of ~9 mg of TP MH showing water loss at 58–85 °C (5 °C/min), **b** tracked $f_{res}$ and $Q$ of a single particle (1200 μm × 85 μm) of TP MH (5 °C/min), **c** DMA thermogram of ~50–100 mg TP MH at 5 °C/min, **d** isothermal TGA thermogram of TP MH (~10 mg) at 50 °C for 70 min, **e** tracked $f_{res}$ and $Q$ during isothermal dehydration of a TP MH resonator (1600 μm × 66 μm) of TP MH at 50 °C for 70 min. The shaded regions in the figures highlight the main dehydration transitions of TP MH.

(crystal) is stiffening. These transitions are proposed to be due to the presence of metastable intermediates, even at 70 °C, as was recently documented using Raman line-focus microscopy[4]. These transitions are also supported by Ostwald's rule of stages that suggests that during recrystallisation the thermodynamically unstable forms appear first followed by the stable form[46].

The DMA thermogram (Fig. 3c) of TP MH shows the main dehydration event between 60 and 90 °C with a maximum tan δ signal 82 °C due to the formation of anhydrous theophylline. DMA shows a larger dehydration range of TP MH compared to TGA and DSC. These results show that although the presented methods detect the main dehydration of TP MH, there are differences in the dehydration temperature ranges of DSC ($\Delta T$ = ~25 °C) and DMA ($\Delta T$ = ~30 °C) in comparison to PMTA ($\Delta T$ = ~11 °C). However, TGA ($\Delta T$ = ~14 °C) and PMTA are comparable. DMA (See Supplementary Figs. 12–15) also shows the changes in the elastic modulus of TP MH during dehydration where a gradual decrease is observed until 60 °C – then an increase appears. The overall differences in the thermal response of TP MH observed with the discussed methods are attributed to dissimilarities in sample pans and sample sizes during TP MH measurements. DSC measurements were performed with the sample in an enclosed pan, TGA in an open pan in a furnace and DMA in a clamped open rectangular sample holder that can cause differences in dehydration due to differences in heat transfer in the measurement geometries.

Figure 3d shows the isothermal dehydration of TP MH (~10 mg), via a two-step weight loss during the first 20 min in the TGA experiment whereas the PMTA thermogram (Fig. 3e) of a single particle of TP MH showed dehydration occurring within

~11 min (See Supplementary Fig. 16 and Supplementary Movie 4). The tracked $f_{res}$ ($\delta f$ = 3.6%) and $Q$ for the isothermal experiment show addition thermal transitions during the dehydration of TP MH (See Supplementary Movie 7) to its stable anhydrous phase. These additional transitions are undetected in the TGA measurement. The differences in the signals from TGA and PMTA measurements are proposed to be due to the thermo-mechanical response of a single particle whereas with TGA only water loss is determined. The fluctuations in $Q$ (50–65 min) shows that the crystal lattice is undergoing a cascade rearrangement[47] throughout the heterogeneous particle resonator as supported by the raw data. Using the same TP MH particle, it was demonstrated that there is a correlation between the thermal transitions and physical bending behaviour of the particle. By analysing optical images obtained from the LDV instrument during dehydration, the movement of the tip of the particle coincided with the thermal transitions observed in the Q for the same particle (Supplementary Fig. 17). A similar effect has previously been seen on thermosalient[48] particles. This can be used as a principle to construct simple characterisation technologies with no need for LDV – just static deflection measurements based on image processing could be performed.

**Repeatability of PMTA and mode analysis.** In order to investigate the repeatability of the PMTA method, eleven measurements (25–90–25 °C, 5 °C/min) on single particles were performed. Table 1 shows the overview of all the measurements performed with these particles and their associated variation (See Supplementary Figs. 18 and 49). There were a range of different shapes (rod, plate) and sizes (length and width) of TP MH

**Table 1 Eleven TP MH particles that were used in the study.**

| TP MH particle | Particle dimensions (μm) | $f_{res}$ (kHz) SD(±) for each particle | PMTA main phase transition (°C) | PMTA average main phase transition (°C) |
|---|---|---|---|---|
| 1 | 1400 × 71 | 15 ± 6.0 | 75–86 | 81 |
| 2 | 1000 × 65 | 19 ± 3.8 | 72–86 | 79 |
| 3 | 1200 × 85 | 23.2 ± 3.87 | 58–82 | 70 |
| 4 | 900 × 42 | 12.3 ± 3.32 | 63–85 | 74 |
| 5 | 1000 × 100 | 24 ± 3.7 | 74–85 | 80 |
| 6 | 530 × 50 | 21 ± 7.2 | 68–84 | 76 |
| 7 | 600 × 50 | 21 ± 3.3 | 76–90 | 83 |
| 8 | 300 × 90 | 68 ± 4.1 | 60–78 | 69 |
| 9 | 400 × 100 | 24 ± 6.2 | 72–82 | 77 |
| 10 | 350 × 80 | 87 ± 5.1 | 68–80 | 74 |
| 11 | 350 × 50 | 88 ± 7.9 | 68–84 | 76 |

The mean width of the particles was calculated to be ~71 μm.

**Table 2 Standard deviation associated with the standard thermal methods used in this study.**

| Particle | DSC main phase transition (°C) SD (±) $n = 3$ | TGA main phase transition (°C) SD (±) $n = 3$ | DMA main phase transition (°C) SD (±) $n = 3$ |
|---|---|---|---|
| TP MH | 73 ± 1.1 | 65 ± 0.8 | 81.8 ± 1.3 |

particles, resulting in differences in the $f_{res}$, where shorter and wider particles such as 350 μm × 80 μm have higher[40] $f_{res}$ in comparison to larger sized particles such as 1000 μm × 65 μm. Table 2 shows the variation associated with DSC, TGA and DMA. As shown in Table 1, with PMTA there is more variation (in comparison to the standard methods) associated with the temperature range at which the dehydration events are taking place. However, the values still fall within the expected range when compared to the bulk methods.

Figure 4a, b shows the thermograms from the particles in Table 1 that were all conducted from 25–90 °C, at 5 °C/min. There is a general trend in the temperature/time range of the main dehydration for the smaller particles versus the larger particles, but not all particles adhere to this trend like 1200 μm × 85 μm particle (faster onset of dehydration in comparison to the other particles at ~1000 μm). The smaller sized particles (plotted with dotted lines) are indicating an earlier onset of dehydration, and this is more apparent with the first particle (300 μm × 90 μm) at 60 °C. The resonator length appears to influence the onset of the linear increase in $f_{res}$ more than the width of the resonator in this study, which is dictated by the anisotropic dehydration of TP MH. This result is also confirmed by Eq. 1. There is an influence of the resonator width as seen in measurements with 400 μm × 100 μm that has a delayed onset of dehydration by ~10 °C in comparison to the particles in the 530 μm × 50 μm range, due to the difference in thickness of 50 μm that influences the loss of water to the boundaries of the particle. A reverse effect is observed with 1200 μm × 85 μm resonator that has a faster onset yet is longer than the 1000 μm long particles. It could be that the narrower width is influencing the dehydration. During the cooling phase, all particles appear to undergo minor shifts in $f_{res}$ with intrinsic dampening effects that are still detectable even during cooling. This can be seen in Supplementary Fig. 19. Another reason for the variations are visible particle defects

(Supplementary Figs. 1–4) that are contributing to the heterogeneity at a macroscopic level.

It has previously been shown that local differences in materials deposited on resonators can be probed by monitoring the resonance response of several vibrational modes[49–51]. Due to this, we tracked two different vibrational modes (15 kHz and 82 kHz) of a TP MH particle (Fig. 5a) during two heating-cooling cycles (25–90–25 °C, 5 °C/min) (Supplementary Figs. 20, 21 and 22), see Fig. 5b–c (for other tracked mode of TP MH resonators see Supplementary Figs. 23 and 24). Based on Eq. 1, the calculated vibrational modes were found to be first and a higher mode (See Supporting Movie 8). The onset of the dehydration (initial frequency drop) appears to start ~2 min earlier for the first mode than for the higher mode at 8 and 10 min, respectively. However, both modes show a $\delta f$ of 21% during the main dehydration event, indicating that this can be explained by a sudden uniform loss of water content that would impact both modes in the same way.

On the first cooling run, $f_{res}$ increases and then decreases upon the second heating cycle. A similar transition trend is observed with the higher mode, but there are minor differences (1–4 kHz) as can be seen in the second heating cycle. For example, at the 35 min mark, the $f_{res}$ of the first mode recovers earlier compared to the higher mode. The second thermal cycle is primarily linked to changes in Young's modulus. The $Q$ of the first mode drops from the start of the experiment (0–6 min). In comparison, the $Q$ for the higher mode initially undergoes a minor drop followed by a major drop during the main dehydration event and appears to maintain a higher $Q$ compared to the first mode throughout the experiment.

The resonance frequency of the first flexural mode has its mass sensitivity maximum at the free end. This explains the fast response of the $f_{res}$ for the first mode as the mass loss occurs in an anisotropic manner from the clamped end and through the crystal. Changes in stiffness and mass of the mechanical structure of the particle are reflected more in the higher modes where more vibrational energy is stored in bending along the particle[52]. In contrast, the first mode has the most mechanical energy stored by bending close to the clamped end. As a result, the $Q$ of the first mode will be very sensitive to changes in internal dissipation at the clamping region. Higher modes will give a more averaged response of both stiffness and internal dissipation through the entire particle. The differences in the thermomechanical responses of the different modes (Supplementary Movies 9–17) demonstrate the added value of tracking multiple modes since more information about the material properties can be extracted (local spatial differences). This can provide an opportunity to examine various interconnected events in different regions (Supplementary Fig. 27) of the particle as they all have different effects on different vibrational modes[53], which is a unique feature of resonating structures.

**Collagen fibres resonator.** Up to this point, an example of a small organic molecule has been presented with elongated particles. In the following example, a thin roll of collagen fibrils was applied as a circular cantilever for thermomechanical characterisation at the particulate level. The first step was to perform bulk analysis with the collagen sample as was done with TP MH. The DSC thermogram (Fig. 6a) shows an endotherm at ~57 °C, which can be attributed to conformational changes occurring within the fibrils before denaturation[54,55]. The broad endotherm between 60–80 °C shows the main transition corresponding to the process of gelatinisation of collagen due to the breaking of internal crosslinks. Another endothermic event at 100 °C was detected and is due to the evaporation of residual water as previously detected[55] (Supplementary Fig. 25 for the DSC thermogram). The TGA

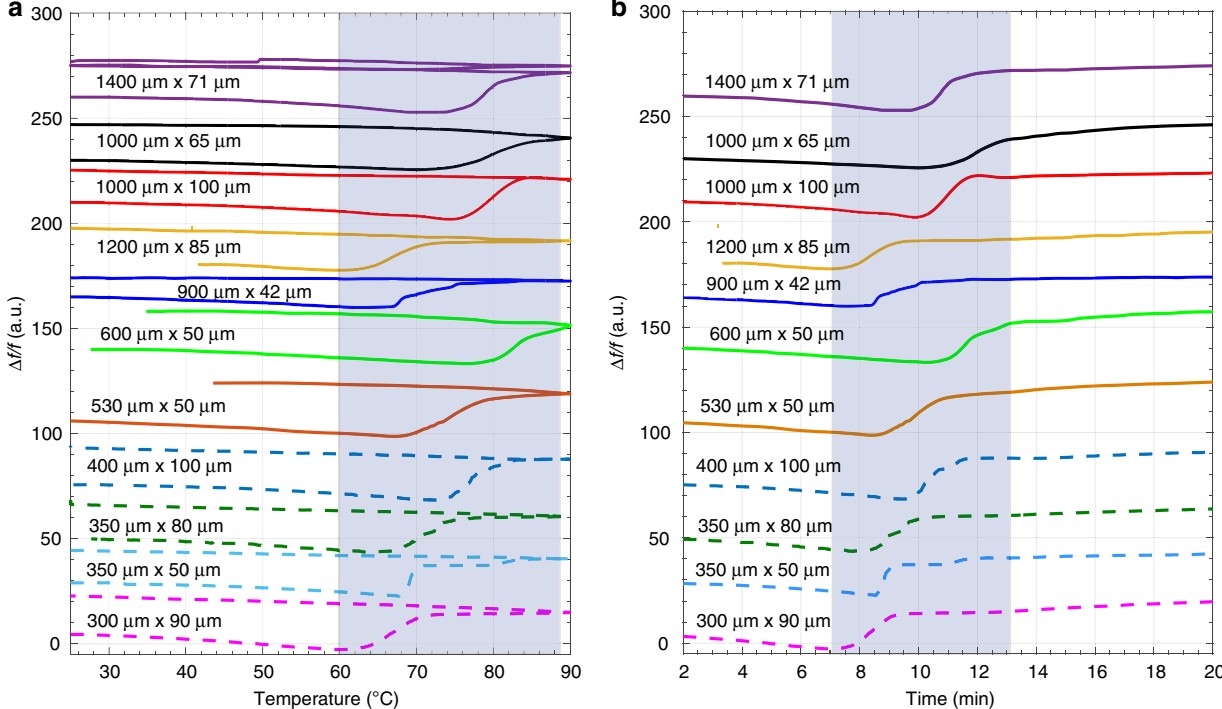

**Fig. 4 Reproducibility of PMTA and analysis of different modes of vibration. a** PMTA thermograms showing dehydration (25–90–25 °C, 5 °C/min) of eleven single particles of TP MH, **b** Time plot of the same eleven particles in Fig. 4a. The dotted lines signify the relatively smaller sized particles in this study. 1400 μm × 71 μm particle was exposed to a second thermal cycle (25–90–25–90–25 °C, 5 °C/min). The shaded regions in the figures highlight the main dehydration transitions of TP MH.

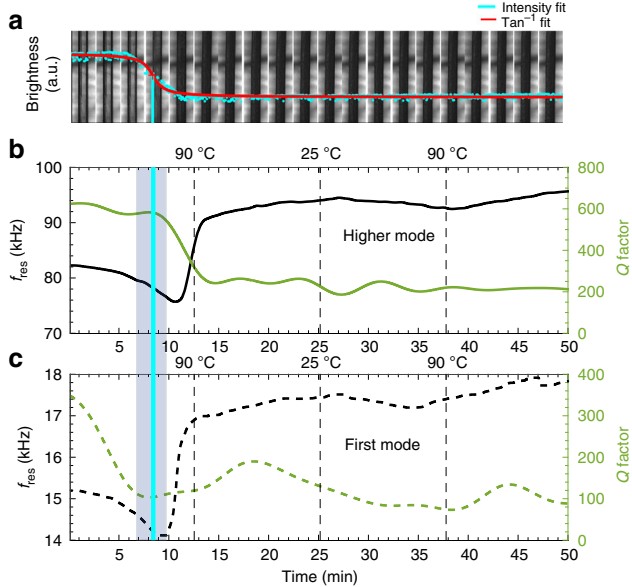

**Fig. 5 Mode analysis of a TP MH cantilever. a** Optical images showing the dehydration (25–90–25–90–25 °C, 5 °C/min), of TP MH particle resonator (1400 μm × 71 μm). **b**, **c** Tracking of two flexural modes of vibration (15 kHz and 82 kHz) of the same TP MH particle during two heating and cooling cycles. The shaded regions in the figures highlight the dehydration transitions of TP MH.

thermogram shows a continuous weight loss occurring during the main transitions (30–200 °C). TGA shows that the compressed sheet undergoes a mass loss of ~13% whereas PMTA records a mass loss of ~8% during the main transition. This difference in mass could be attributed to a mass loss prior to analysis. Figure 6b

shows a roll of collagen compressed sheet. Figure 6c shows a small piece (~80 μg) from the external layer that was manually extracted using tweezers, rolled up and applied as a cantilever (circular cross-section). A zoom-in of the collagen fibrous structure can be seen in Fig. 6b3. It is worth noting that collagen fibrous structures are formed by collagen fibrils, wherein the collagen fibrils consist of tropocollagen molecules that assemble in a staggered pattern[56]. Visually, the fibrils have the appearance that is characteristic of collagen with an average diameter of 50 nm (in agreement with the gelation procedure applied)[57], and can be modelled as five tropocollagen molecules staggered side-by-side with an offset of $D = 67$ nm between two neighbours[58] (Supplementary Fig. 26 shows the TEM image of the collagen fibrils). The fibrous structure does have a varying width from the clamped end to the end of the scanned area using PMTA.

The PMTA thermogram in Fig. 6f shows a linear drop in $f_{res}$ (See Supplementary Movie 18) that is inferred to be the conformational change of collagen that is detected by DSC. The $f_{res}$ drops ($\Delta f = \sim 2.03\%$) gradually during the denaturation (48–75 °C) of the collagen fibrils due to the thermal expansion of the fibrils. From 75 °C, there is a gradual increase in $f_{res}$ up to 88 °C, followed by a sharp increase in the signal from 88–100 °C, then reaches a plateau until 110 °C. This is followed by a subtle drop in $f_{res}$ then an increase from 120 °C via multiple minor thermal transitions up to 200 °C. The increase in $f_{res}$ is attributed to thermal stress and thereby increase in the density of collagen. The Q increases from 30 °C then decreases during the main denaturation of the collagen fibrils. From 80–140 °C, there is a subtle increase in Q followed by its recovery at 140–200 °C in multiple steps. The denaturation of collagen fibres has previously been found to be a result of the unravelling of the triple helix structure[59] that is coupled with the breakage of hydrogen bonding (See Supplementary Movie 19)[60]. In comparison to

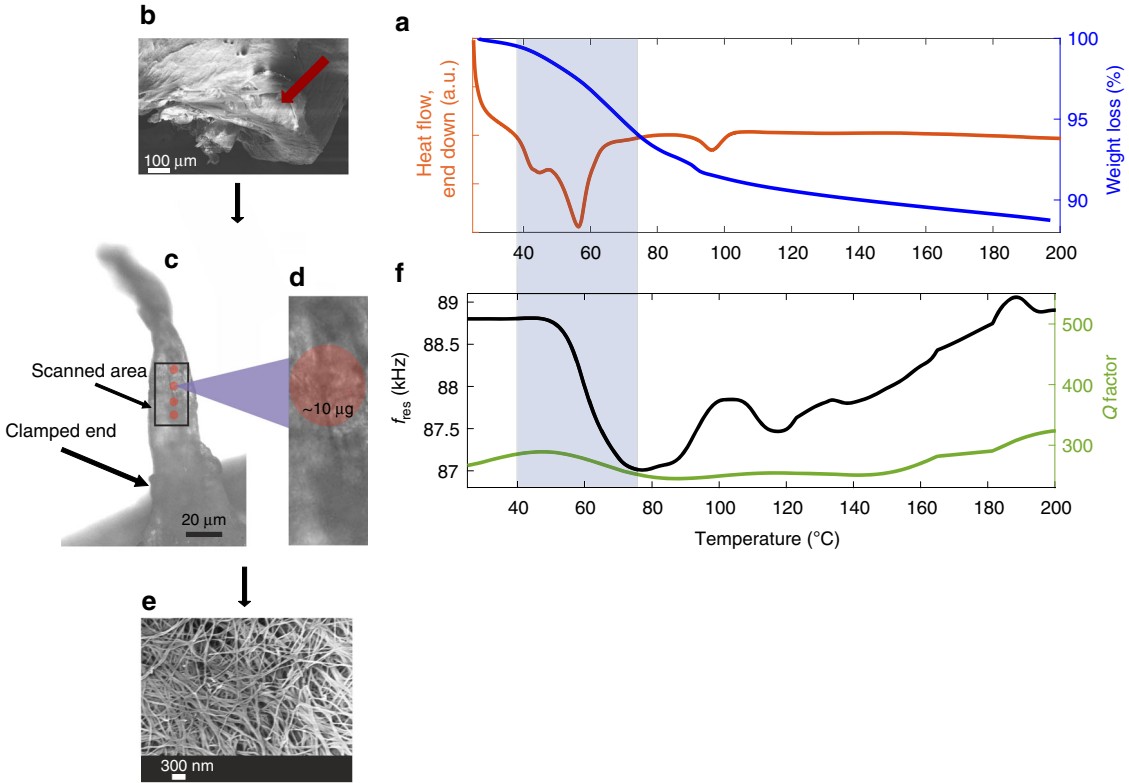

**Fig. 6 Thermomechanical analysis of collagen fibres. a** DSC and TGA thermograms (25–200 °C, 20 °C/min) of collagen fibres (~14 mg). **b** Scanning electron microscopy (SEM) image shows a bundle of collagen including several layers, where a small piece of the external layer (shown by the red arrow) is cut and used for the PMTA measurement. **c** A representation of the area that was scanned during the PMTA experiment and the thin roll of rat-tail collagen fibrils that was applied as a cantilever. **d** A zoom in of **c** for visualisation of the entanglement of the collagen fibrils. **e** A zoom in of the collagen fibrils from **b2 f** PMTA thermogram of collagen fibres (25–200 °C, 20 °C/min). The shaded regions in the figures highlight the main transitions of the collagen fibres.

DSC and TGA, PMTA shows additional thermal transitions that are related to the denaturation of collagen fibrils[38,56].

## Discussion

The established thermal methods yield useful results for both small organic molecules and macromolecules. However, in drug discovery, when only a limited amount of material is available (including pure), it can be challenging to use these methods since a minimum of milligram amounts of material is required for analysis[61]. PMTA has a number of advantages of resonant NEMS and MEMS systems in comparison to the standard bulk techniques for thermomechanical analysis as well as the established NEMS/MEMS-based techniques. In our studies, even a single PMTA measurement coincides with the expected main phase transitions detected by the bulk methods – however also revealing additional transitions. These transitions are very challenging to decipher with the bulk methods because they relate to the intraparticulate changes and inter-particulate differences (to a lesser extent). This is due to the noted differences between particles and an averaged bulk response from the standard methods. With increasing drug characterisation studies being performed on small sample sizes it is important to consider these differences–otherwise this can lead to a misrepresentation of the bulk behaviour of the sample.

The arrangement of molecules in a crystal typically does not allow for perfect crystalline patterns (Supplementary Fig. 5b, c) in organic materials primarily due to its growth during recrystallisation in a solution that results in varying particle sizes, shapes, as well as defect densities[62].

In comparison to optical cavity resonators (Qs in the millions), the Q values in these studies are significantly lower. The Q is primarily influenced by the intrinsic dampening and anchor clamping of the single particles (See Supplementary Figs. 28 and 29). Due to the micron-sized particles that were used, the dominating influences in the Q were from intrinsic dampening as water is initially present in the crystal lattice. Overall, the Q values (~≥100) achieved in these studies were acceptable for achieving reliable resonance frequency detection for the measurements.

The temperature of the freestanding particle needs to be considered. This can vary due to a number of effects including convective heat transfer to the surrounding air and cooling from evaporating water molecules. Finite element method (FEM) simulations (Supplementary Figs. 28–43) that the difference in temperature of the stage and the particle does not deviate more than ~0.3 °C (at a stage temperature of 70 °C). PMTA can be easily adapted to be portable by using cheaper optical or electronic readout options, which makes it usable in different research, development and manufacturing phases in materials science[63].

Overall, PMTA is able to measure on a single particle that can be picked from a batch of particles, has millisecond time resolution for investigating short-lived forms, and avoids the need for fabricated resonators (such as silicon nitride strings/membranes/cantilevers), that may interfere with the measurements of the deposited sample during thermomechanical analysis[64]. When using fabricated resonators adhesive and contact mechanics of the measured sample needs to be accounted for. This is known to be affected by surface roughness[65]. Whereas with PMTA this effect is essentially eliminated as we directly measure on the material itself. The take away from comparing a single particle to the bulk

methods is that there can be additional thermal transitions that are not "washed out".

In summary, we presented the first examples of using single particles of small organic molecules as well as macromolecules based particles as resonators in materials science. Our findings show the complexity of the thermomechanical response of particles during their phase transitions. PMTA provides new insights into thermal transitions of the studied materials in comparison to standard thermal methods. Furthermore, additional information on changes in mechanical properties during thermal cycling is gained. This could be useful information in the early stages of material research since usually only very small amounts of materials (nano- microgram) are available and there is an increasing interest in understanding the fundamental behaviour of small and large molecules very early in the development process to avoid appear- ance of unexpected phase transitions during material development, manufacturing or storage. In terms of predictability of the physio- chemical and mechanical properties in early phase development of a material, it would be ideal to take multiple individual measure- ments and compare the average to a standard bulk method as was demonstrated in this study. Ideally, PMTA can be used with complementary methods such as variable temperature XRPD or Raman spectroscopy to explain the nature of the observed phase changes in materials. This method offers a new way of performing thermomechanical measurements in NEMS/MEMS, pharmaceu- tical research and biomedical sciences. Finally, the possibility of monitoring unique features of individual particles might open up for entirely new fundamental studies on e.g. influence of particle shape and defects on thermomechanical behaviour.

## Methods

### Materials and methods

*Recrystallisation of TP MH particles*. Anhydrous (AH) theophylline (TP) was purchased from Sigma-Aldrich (See DSC thermogram in Supplementary Fig. 44) and recrystallised in distilled water using the cooling evaporation crystallisation method to form TP MH. TP MH was prepared by dissolving anhydrous theo- phylline in distilled water (60 mg/mL, 80 °C) then allowed to slowly cool to room temperature and needle/rod shaped particles with lengths between 500 and 2000 μm, and widths between 10 and 200 μm) of TP MH formed. The sizes of the particles used were between 300 and 1800 μm in length and 30–110 μm in width. Standard thermal methods confirmed the solid-state forms of TP (See Supple- mentary Figs. 45–46); TP AH form II (CSD ref code: BAPLOT01[66]), and TP MH (CSD ref code: THEOPH01)[67]. Freshly prepared samples were used to conduct all the DSC, TGA, DMA and PMTA experiments. Experiments were also conducted on the recrystallised anhydrous theophylline, see Supplementary Figs. 47 and 48 and Supplementary Movie 20.

*Preparation of collagen fibres*. The protein model (rat-tail collagen type I) was bought from First Link Ltd, West Midlands, UK. Material verification was per- formed using XRPD, DSC and TGA (Fig. 6 and Supplementary Fig. 49).

A collagen fibrous sheet was prepared using the technique of collagen hydrogel compression as described previously[68]. In brief, 4 mL of sterile rat-tail collagen type I solution (2.06 mg/mL protein in 0.6% acetic acid; First Link Ltd, West Midlands, UK) was mixed with 0.5 mL 10X Eagle's minimum essential medium (MEM; First Link Ltd.). The solution was neutralised with 2.5 M NaOH, after which 0.5 mL of alpha-MEM medium was added. The 4 mL collagen solution was cast into circular- shaped moulds (diameter of 34 mm) and was incubated at 37 °C for 20 min to undergo the gelation procedure. Then, the gelled construct was transferred onto blotting elements, consisting of a layer of sterile 110-mm-thick nylon mesh (~40 mm mesh size) and a sterile 400-mm-thick stainless steel mesh (mesh size ~200 mm), which were placed on top of three sterile gauze pads. The set gel was then covered with a second nylon mesh and a loading plate (as a static weight) (120 g) for 5 min at room temperature[69], leading to the formation of a thin sheet of collagen. To facilitate handling, the collagen sheet was rolled, kept hydrated inside alpha-MEM culture media and stored at 4 °C in the fridge after preparation. For the thermomechanical analysis, a very small piece of the external layer of the roll (including bundles of collagen fibrils) was cut (Fig. 6b–e), washed thoroughly with milli-Q water, and the excess water was drained away. Finally, the sample mass was measured then used for solid-state confirmation then PMTA measurements.

Compression of collagen hydrogel has the advantage of alleviating the need to use cross-linker whilst making a sheet of collagen fibrils with a minimum level of entanglement between adjacent fibrils (Fig. 6e). Since entanglements impose topological constraints on polymer conformations[70], no use of chemical

crosslinking has reduced the level of intra-chain entanglements. By applying a mechanical compression force the water from the hydrogel structure is removed and provides us with a bundle of fibrils, very similar to the collagen structure in native tissue. By this procedure, a limited amount of manipulation was required and facilitated fast sample preparation.

*X-ray powder diffraction*. A PANalytical X'pert PRO X-Ray Diffractometer (PANalytical B.V., Almelo, Netherlands) equipped with a θ/θ goniometer and a solid-state PIXcel detector was used. Nickel-filtered CuKα (λ = 1.5418 Å) radiation was generated at a tube voltage of (45 kV) and current (40 mA), respectively. The samples were scanned in reflection mode between 5.01° and 30.0° with a scan speed of 0.0673° 2θ and the step size of 0.0263° 2θ. The data were analysed using the X'Pert Data Collector software (PANalytical, Almelo, Netherlands). Measurements were done in triplicate.

### Thermal analysis

*Differential scanning calorimetry*. A Discovery DSC (TA Instruments-Waters LLC, New Castle, USA) was used to perform the DSC measurements and this instrument is controlled by TRIOS software (TA Instruments, New Castle, DE, USA). The theophylline and collagen samples (~14 mg) were placed into T-zero aluminium pans and sealed using a Perkin-Elmer crimper (holes in the lid). The samples were heated based on the expected dehydration or denaturation temperature ranges. The samples were heated at a controlled heating rate of 5 and 20 °C/min under nitrogen, purge (40 mL/min). Measurements were done in triplicate.

*Thermal gravimetric analysis*. The TGA thermograms were obtained using a Dis- covery TGA (TA Instruments, New Castle, DE, USA), which was controlled by TRIOS software (TA Instruments, New Castle, DE, USA). In all, 9–13 mg of TP MH were weighed in a platinum pan using a microbalance under a nitrogen purge (40 mL/min). The TP MH isothermal measurements were performed at 50 °C for 70 min. The TP MH and collagen samples were heated between 25–250 °C (5 °C or 20 °C/min). Measurements were done in triplicate.

*Dynamic mechanical analysis*. In total, 50–100 mg of TP MH was loaded onto a stainless steel powder pocket sample holder. The sample was clamped into a 35 mm dual cantilever clamp of a DMA Q800 (TA Instruments- Waters, New Castle, DE). Measurements were performed in a multifrequency-strain mode using a frequency of 1 Hz, an amplitude of 20 μm and a heating rate of 5 °C/min from 25–100 °C. Data analysis were performed by TA Universal Analysis software. Measurements were done in triplicate.

*Laser Doppler vibrometer*. The thermomechanical spectra were measured using a laser Dopper vibrometer (MSA-500 from Polytec GmbH, Germany) in air with a 633-nm laser beam (~3 μm, FWHM). In an experiment, the laser spot on a particle was only active on a given spot on the measurement grid for no more than a tenth of a second for the majority of particles, thereby eliminating the effect of laser- induced dehydration (Supplementary Figs. 34, 35 and 37). The LDV is fitted with a microscope that collects images of the particle after each scan of the defined measurement grid on a particle has been completed. A piezoelectric element (NAC6024, Noliac A/S, Kvistaard, Denmark) was used for actuation. The vib- rometer was on a pressurised optical table while conducting all measurements in order to limit external vibrations. A 5x microscope objective was used for all measurements.

*Particle mechanical thermal analysis measurement setup*. In a fresh batch of crys- talline particles, a small sample was placed on a glass slide and a single particle was manually chosen using a polarised light microscope (PLM). The sample holder was then placed on a glass slide and the chosen particle was placed on an Aluminium (Al) sample holder whilst using the PLM for visual aid in placing the particles.

Al sample holders were designed using AutoDesk Inventor Professional 2017 and milled using a 0.2 mm end mill. A schematic of the measurement setup is shown in Fig. 1a, where a single particle of the material was mounted on an Al block and clamped (one end) on this block (Fig. 1b) using an Ultraviolet activated glue. By using UV activated glue, it is possible to begin measurements after exposing the glue to the UV rays for only a few seconds. The sample holder with the clamped particle(s) was then placed on the Linkam hot stage (FTIR600 stage, T95 Linksys controller) for temperature control. A piezoelectric crystal was used for actuation of the model particle(s) and placed adjacent to the Al holder. The LDV has its own computer that also consists of a junction box and controller. The defined grid on the particle(s) being measured was used to visualise the resonant vibrations with a Polytec vibrometer software (PSV 14.2). The resonance frequency is defined as the maximum amplitude of vibration for a given material. In addition, theoretical calculations of the resonance frequencies for the particle(s) were also performed using Eq. 1. The model particles were characterised by continuously measuring their thermomechanical spectra during thermal cycling and subsequent data analysis. All 15 particles were measured in the air.

*PMTA: data analysis*. All the data generated by the LDV was analysed using MatLab R2017a (Supplementary Figs. 50–52) for more a more detailed explanation

of the data analysis steps. In order to process the image of the particle, an area of the particle is defined manually by drawing a square (Supplementary Movie 21). An arctan function is then fit to the mean of the pixel intensity values of the particle over the time profile of the experiment. The points with a too low SNR were manually excluded. The data were imported into MATLAB using a customised script that tracks the $f_{res}$ through quasi-Lorentzian fits to the thermo-mechanical spectra. To measure the $Q$, this parameter was extracted from a fit to the phase signal around the $f_{res}$. Finally, a low-pass filter was applied to both the $f_{res}$ and $Q$ data to reduce the noise (Fig. 2b).

## Data availability

The data that supports the findings in this study are available within the paper and the supplementary Information. Additional data are available from the corresponding author upon reasonable request.

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

## Acknowledgements

We would like to acknowledge the Center for Intelligent Drug Delivery and Sensing Using Microcontainers and Nanomechanics (IDUN) funded by the Danish National Research Foundation (grant no. DNRF122), Velux Foundations (grant no. 9301) and Novo Nordisk Foundation (grant no. NNF17OC0026910) for the funding of this project. The DMA measurements were supported by NordForsk for the Nordic University Hub project #85352 (Nordic POP, Patient Oriented Products). The authors also thank Anders Carlsen for the FEM simulations.

## Author contributions

P.O.O. carried out all PMTA experiments, wrote scripts for analysing the data, performed the data analysis and wrote the paper. P.E.L. wrote scripts for analysing the data, supported the data analysis and reviewed the manuscript. E.O.K. performed the DMA measurement and analysed the DMA data. F.A. prepared the collagen sheets, performed the electron microscopy imaging of the collagen samples. P.E.L, A.B., J.R. and T.R. supervised the project, reviewed the manuscript and guided in the discussion of the results.

## Competing interests

The authors declare no competing interests.
