## [Peer Review File · Nature Communications]

Reviewers' comments:

Reviewer #1 (Remarks to the Author):

The presented draft 'Single Particles as Resonators for Thermomechanical Analysis' it's genuinely interested and new and should be considered for publication after major revision.

My comments:

In the introduction part, authors should discuss in more detail the difference between the presented, resonator, and MEMS/ NEMS research. At the moment, it's a little ambiguous since the given system is not an optical cavity.

Authors nicely defined the quality factor as the measure of mechanical damping but the similarity and difference with the optical cavity quality factor should be additionally discussed and highlighted.

Answer: We thank the reviewer for raising this point and we agree that the given systems investigated in this manuscript are not an optical cavity. We have added this discussion in the revised manuscript.

The standard material that is used in order to fabricate resonators is normally silicon nitride and as such, they have significantly higher mechanical Qs in comparison to resonators used in our studies. Thermomechanical measurements using silicon nitride based resonators are normally carried out in vacuum in order to avoid the dampening effects of air. In our measurements, the main limiting factor for the mechanical Qs is the volumetric losses due to the presence of water in the TP MH particles as well as the large-sized particles that were used in the studies ranging from 300 -1400 μm in length to 42 to 100 μm in width. These are considered relatively large particles even in MEMS and this has now been further discussed in the manuscript in the results and discussion section. In general, we obtained Qs generally above 100, which is acceptable for our experiments and MEMS research when working with cantilevers.

Due to the transition of the theophylline monohydrate (TP MH) particles, we do see a visible change in the opacity of the particle over time and this is highlighted in Figure 2 as well as Supplementary videos 1-6. In addition to this, we have also added plots in the supplementary information showing the changes in the opacity in different areas/boundaries of the particle during dehydration (See Supplementary Figure 27, page 25).

Figures 3 and 4 are the most exciting and essential for the draft. The major advantage of the presented method is the possibility to achieve thermomechanical characterization at the single-particle level.

The given difference between bulk TGA and DSC characterization and measurement at the single-particle level should also discuss in detail taking into account the difference between the nanoscopic and mesoscopic structure of polymers.

Answer: Understanding the fascinating and complex structure and dynamics of polymeric materials has been an ongoing challenge for many decades. From the point of view of molecular simulations, the spectrum of length and time scales associated with polymer melts of long chains poses a formidable challenge to studying their long-time dynamics.^{1,2} The topological constraints arising from chain connectivity and uncrossability (entanglements) dominate intermediate and long-time relaxation² and transport phenomena when polymers become sufficiently long. Atomistic molecular simulations of dense phases of soft matter prove to be difficult for many systems across length and time scales of practical interest. Even coarse-grained particle-based simulation methods may not be applicable due to the lack of faithful descriptions of polymer–polymer and polymer–surface interactions. Since complex interactions between constituent phases at the atomic level ultimately manifest themselves in macroscopic properties, a broad range of length and time scales must be addressed and a combination of modelling techniques is therefore required to simulate meaningfully the bulk-level behaviour of nanocomposites.³

References

1. Brown, R. a., Wiseman, M., Chuo, C.-B., Cheema, U. & Nazhat, S. N. Ultrarapid Engineering of Biomimetic Materials and Tissues: Fabrication of Nano- and Microstructures by Plastic Compression. *Adv. Funct. Mater.* **15**, 1762–1770 (2005).
2. Everaers, R. *et al.* Rheology and Microscopic Topology of Entangled Polymeric Liquids. *Science* **303**, (2004).
3. Vogiatzis, G. G. & Theodorou, D. N. Multiscale Molecular Simulations of Polymer-Matrix Nanocomposites or What Molecular Simulations Have Taught us About the Fascinating Nanoworld. *25*, 591–645 (2018).

How do the ordering of polymers and the lack of significant entanglement of polymer chains at the single-particle level affect thermomechanical properties?

Regarding the effect of “lack of significant entanglement of polymer chains at the single particle level on thermomechanical properties”, we would like to attract the attention of the reviewer to below-mentioned points:

Answer: We have applied compressed collagen construct as the polymeric model of a protein in this study. With regard to the main aim of this system, i.e., using a much lower amount of the material for a comprehensive thermomechanical study, a tiny piece of the sheet was cut and used. To minimize the level of polymer chain entanglements, we have considered avoiding the use of cross-linkers to solidify the protein hydrogel. Instead, physical compression was used to remove the aqueous microenvironment surrounding the hydrophilic polymer chains to make the hydrogel¹. Indeed the compressed collagen sheet is a nice example of a polymeric structure with a lower amount of entanglements compared with other conventional collagen-based hydrogels. Since entanglements impose topological constraints on polymer conformations², no use of chemical crosslinking has reduced the level of intra-chain entanglements. However, we cannot consider the polymeric sample in the level of “single particle”, from the molecular simulations point of view. In fact, polymers include long chains, where chain connectivity and uncrossability (entanglements) dominate intermediate and long-time relaxations in polymer melts³. It consequently poses challenges to study the long-time dynamics of the polymers, and any proposed method such as PMTA that is providing the possibility of using microgram amounts of material, whilst maintaining comprehensive data would be of great value.

Does the presented method allow us to investigate the property of the single polymer chain almost not accessible by other methods?

More deep physical discussion is needed for the journal like Nature Comm., and I encourage authors to provide deeper discussion in the revised version in order to make their case stronger.

Answer: Thank you for this important point. This has now been addressed on the entire text of the revised manuscript.

In Figures 2 and 4, the error of measurement should be presented, and the estimation of the error should be described and discussed in the main text.

Answer: The authors have now addressed this in Figure 2, 4, Table 1 and 2 of the revised manuscript.

Taking into account the importance of investigation at the single-particle level, which reveals the structure-

function relationship at the nanoscale, the novelty of the presented draft is truly high. I consider the presented draft as a significant step in better understanding of soft mechanics at the nanoscale.

Also, the presented method could have a lot of applications in biotechnology and pharmacy, and in other domains where thermo-mechanical characterization is essential.

In the end, I am recommending the major revision of this draft.

Reviewer #2 (Remarks to the Author):

This work proposes the thermal analysis of single microparticles directly through mechanical characterization by using Laser Doppler Vibrometry, a broadly used technique. This approach is very original and elegantly simple, so it could be adopted by a large number of laboratories. The proposed method could have a large impact as a simple, affordable and robust technique for characterization of drugs, both in the development and manufacturing phases. Given this high potential impact, I advice publication in Nature communications, in case some critical questions can be clarified.

My most relevant concern about this work is that the number of experiments performed is very low. Only three different particles have been measured for TP MH. This is a very low number of experiments to demonstrate a new method, particularly when these three experiments show significant variations. Although bulk methods are also not perfectly reproducible, this does not justify the low number of experiments. Moreover, the tests do not seem extremely complicated; otherwise the technique would not be of use to the cited applications in drug development. More experiments performed on a larger number of single particles, together analysis of particle geometry and size for each experiment needs to be done.

Answer: We thank you for this critical comment in the advancement of our work. We now added 15 data sets that include the analysis of 11 TP MH particles. We have presented these findings in Table 1, and Figure 4. These finds have also been discussed in the main text of the revised manuscript as well as the addition of Q data in Supplementary Figure 19 (page 18) & Supplementary Movies (1-6) for visualisation purposes.

The transitions at 45 C and 70 C observed only by PMTA should be further investigated. Page 6, lines 97-104. This paragraph is speculative. The presence of the cited metastable phases needs to be demonstrated with alternative techniques, as Raman spectroscopy.

Answer: The transitions observed at 45 C and 70 C has now been explained in the revised manuscript page 5 and Supplementary Figure 9 as well as Supplementary Note 2 (page 8-9) on the presence of the metastable forms. In order to supplement this description, we discuss the points below raised by the reviewer further.

Figure 1. a) Optical images showing the dehydration of TP MH to its anhydrous form b) concentration profile of TP MH to TP AH form II c) chemical maps of TP MH (left map, red colour), TP MS (middle map, green colour) and TP AH form II (right map, blue colour) d-f) Raman spectra for TP MH, TP MS and TP AH form II respectively. *This figure was obtained from the Supplementary Information of Okeyo et al (2019).*⁴

This is an important point that has been raised by the reviewer. In our previous publication entitled, “Imaging of dehydration in particulate matter using Raman line-focus microscopy”⁴, we carry out similar thermal measurements at the single-particle level using theophylline monohydrate (TP MH) from the same batch as in this study. While tracking the dehydration of TP MH using Raman line-focus microscopy, it was possible to spatially resolve the presence of multiple solid-state forms of this drug. We found that metastable forms appear in different regions of the particle during dehydration (25 - 90 °C, 10 °C/min). In Figure 1 the optical images showing the dehydration of TP MH through the formation of dark regions. The concentration profiles and chemical maps (Figure 1b,c) show that at 45 °C TP MH is mainly present and at 70 °C the metastable form of theophylline is present and there is subsequently a transformation to the stable anhydrous form. The hyperspectral Raman results are supporting the internal dampening that is occurring in the particle during dehydration. In addition to this, we performed variable temperature x-ray powder diffraction (VT-XRPD) and showed that there are related changes to the unit cell dimensions that are occurring at these temperature points.

This is why it is proposed that the changes in the quality factor at 45 and 70 °C are due to the rearrangement of the crystal lattice. Using molecule dynamics such changes were also captured showing the dehydration of TP MH to its stable anhydrous form by Larsen et al (2019)⁵. In this work, they found that this process occurs via two forms. The metastable form of TP has been documented by multiple groups.^{6,7,8}

References

4. Okeyo, P. O. *et al.* Imaging of dehydration in particulate matter using Raman line-focus microscopy. *Sci. Rep.* **9**, 7525 (2019).
5. Larsen, A. S. *et al.* Determining short-lived solid forms during phase transformations using molecular dynamics. *CrystEngComm* **21**, 4020–4024 (2019).
6. Phadnis, N. V. & Suryanarayanan, R. Polymorphism in Anhydrous Theophylline—Implications on the Dissolution Rate of Theophylline Tablets. *J. Pharm. Sci.* **86**, 1256–1263 (1997).
7. Airaksinen, S., Karjalainen, M., Räsänen, E., Rantanen, J. & Yliruusi, J. Comparison of the effects of two drying methods on polymorphism of theophylline. *Int. J. Pharm.* **276**, 129–141 (2004).
8. Nunes, C., Mahendrasingam, A. & Suryanarayanan, R. Investigation of the Multi-Step Dehydration Reaction of Theophylline Monohydrate Using 2-Dimensional Powder X-ray Diffractometry. *Pharm. Res.* **23**, 2393–2404 (2006).

A concern of lower importance is that the resonance frequency and Q analysis is based on considering the microparticles as beams (Equation 1). In this work, only elongated particles are tested, so this is acceptable; but this analysis limits the applicability of the method to this single geometry. Since the method could also be applied to particles not shaped as beams; as the theoretical treatment could be generalized to any harmonic oscillator, the authors may choose to discuss this in the main text.

Answer: We appreciate the reviewer pointing out this important point. The authors have addressed this in the revised manuscript in the introduction by discussing the generalizability of our method (page 2 of the main text of the revised manuscript) as well as highlighting the application of the collagen resonator with a circular geometry in comparison to the TP MH particles. The TP MH particles also have varying geometries that can be seen through the optical images of the particles measured in our study in Supplementary Figures 1-4.

I will now list some minor issues I have found in the manuscript.

Answer: The authors are thankful for the issues raised below by the reviewer and each individual issue has now been addressed in the revised manuscript with relevant cited articles.

-A more extensive review and citation of works related to the application of DSC, TGA and DMA to drug development is necessary (page 1 lines 18, 19)

-The need for characterization methods of thermal transitions with small sample volumes should be further discussed and relevant papers cited.

-Page 2, line 29, text: MEMs for pharmaceutical research, the reference (12) is about frequency modulation AFM and magnetic transitions on thin films, unrelated to the text, please cite relevant papers about MEMS for pharmaceutical research.

-Page 2, line 35, the interest in investigating short lived phases is cited but not discussed, neither related papers are cited, the relevance of the millisecond time resolution of this new method should be discussed.

-Line 36, how alternative methods, as string resonators, may interfere with the measurements should be further explained or relevant literature cited.

-Line 37, variation in data from single particles is cited. This is clearly a disadvantage of the present method, discuss about this.

-Line 45, check misspelling

-Equation (1). The geometry and size of the particles studied here should be introduced as early as page 3, before equation (1) is discussed.

Reviewers' comments:

Reviewer #1 (Remarks to the Author):

In the revised draft and the rebuttal letter, the authors provide detailed answers on all referee questions.

I am fully satisfied with their answers, and the revised draft is significantly improved from the previous version.

Taking this fact into account, as well as the novelty and potential impact of the presented research, I am recommending the acceptance of this draft in the unchanged form.

Reviewer #2 (Remarks to the Author):

The authors have answered all the queries very satisfactorily. The paper has improved considerably in completeness and clarity. I consider this work is suitable for publication as it is now.